# GraphPhos: Predict Protein-Phosphorylation Sites Based on Graph Neural Networks

**DOI:** 10.3390/ijms26030941

**Published:** 2025-01-23

**Authors:** Zeyu Wang, Xiaoli Yang, Songye Gao, Yanchun Liang, Xiaohu Shi

**Affiliations:** 1College of Computer Science and Technology, Jilin University, Changchun 130012, China; wangzeyu21@mails.jlu.edu.cn (Z.W.); xiaoliy22@mails.jlu.edu.cn (X.Y.); sygao23@mails.jlu.edu.cn (S.G.); ycliang@jlu.edu.cn (Y.L.); 2School of Computer Science, Zhuhai College of Science and Technology, Zhuhai 519041, China

**Keywords:** graph neural network, post-translational modification, prediction of protein-phosphorylation sites

## Abstract

Phosphorylation is one of the most common protein post-translational modifications. The identification of phosphorylation sites serves as the cornerstone for protein-phosphorylation-related research. This paper proposes a protein-phosphorylation site-prediction model based on graph neural networks named GraphPhos, which combines sequence features with structure features. Sequence features are derived from manual extraction and the calculation of protein pre-trained language models, and the structure feature is the secondary structure contact map calculated from protein tertiary structure. These features are then innovatively applied to graph neural networks. By inputting the features of the entire protein sequence and its contact graph, GraphPhos achieves the goal of predicting phosphorylation sites along the entire protein. Experimental results indicate that GraphPhos improves the accuracy of serine, threonine, and tyrosine site prediction by at least 8%, 15%, and 12%, respectively, exhibiting an average 7% improvement in accuracy compared to individual amino acid category prediction models.

## 1. Introduction

Protein post-translational modifications (PTMs) refer to the covalent modifications that occur after protein translation and play an extremely important role in various cellular functions and biological processes [1,2]. More than 200 different post-translational modifications have been discovered [3], mainly including phosphorylation, glycosylation, acetylation, ubiquitination, carboxylation, ribosylation, and methylation. Different types of protein PTMs occur at specific positions within the amino acid sequence. Phosphorylation can activate, deactivate, or modify protein functions by altering protein structure and conformation, such as signal transduction pathways, metabolic activities, regulation of protein functions, and cell apoptosis [4,5,6]. Studies have shown that some diseases, such as cancer, diabetes, and neurodegenerative diseases, are caused by abnormal phosphorylation [7,8].

Phosphorylation is one of the most common protein PTMs, which involves the binding of a phosphate group to specific amino acid residues, primarily concentrated on serine, threonine, and tyrosine residues, with a small amount occurring on arginine, lysine, and histidine residues [9]. Phosphorylation sites refer to the amino acid residues that bind to phosphate groups during protein-phosphorylation reactions. The identification of phosphorylation sites is fundamental to conducting research on protein phosphorylation. Currently, the biological experimental methods used to determine phosphorylation sites generally include site-directed mutagenesis, mass spectrometry [10], proximity labeling assays [11], immunoprecipitation [12], and others. However, these experimental techniques are costly in terms of labor and time and often result in false results. Many PTMs remain unidentified or are misclassified, and they overlook the relevant mechanisms of protein PTMs involved in cellular and biological processes [13]. Computational methods can reduce experimental costs, improve prediction speed, and accelerate research progress, making them effective approaches for identifying phosphorylation sites.

Methods based on deep learning possess excellent feature extraction capabilities that can overcome the limitations of traditional machine learning, offering advantages of higher efficiency, accuracy, and scalability. When combined with manual feature engineering, deep-learning methods can describe the biological properties of proteins more comprehensively.

Prediction of protein-phosphorylation sites can be categorized into general and kinase-specific phosphorylation site-prediction methods based on whether differences in kinases play a catalytic role in the phosphorylation process. This paper focuses on general phosphorylation site prediction.

General phosphorylation site-prediction methods can be roughly categorized into four types according to different approaches: identification methods based on simple consensus patterns, clustering methods based on sequence similarity, traditional machine-learning methods, and deep-learning methods. The first type of method looks for specific positions that characterize the amino acid pattern of phosphorylation. For example, the motif discovery algorithm Motif-All [14] uses support constraints to predict phosphorylation motifs; the C-Motif [15] model is used to identify effective phosphorylation motifs without redundancy under prior knowledge. Additionally, methods like ELM [16], PROSITE [17], and HPRD [18] belong to this category. However, these methods rely on the presence of precise motifs around phosphorylation sites. The second category, based on sequence similarity, aims to provide higher scores to query peptides with higher similarity to known phosphorylated peptides using sequence similarity measures such as BLOSUM62 [19] matrix. Algorithms like PostMod [20] and PSEA [21] fall into this category. Both of these types of algorithms are typically based on limited data, thus having significant limitations in predicting phosphorylation sites.

Traditional machine-learning methods integrate multiple features and can improve prediction capabilities. Over the past decade, machine-learning methods have been widely applied in the field of phosphorylation site identification. Several methods use artificial neural networks to predict protein-phosphorylation sites, such as NetPhos 3.1 [22], AMS 3.0 [23], etc. The classic algorithm using hidden Markov models is KinasePhos [24]; Commonly used methods employing support vector machines include Musite [25], PhosphoSVM [26], PredPhosph [27] and PPRED [28], etc. Models based on BDT and CRF, such as PPSP [29] and CRPhos [30]. PhosPred-RF [31] combines sequence information with the random forest algorithm to predict phosphorylation sites by capturing the differential characteristics between phosphorylation and non-phosphorylation sites, iPhos-PseEn [32] integrates four different pseudo component analysis features into an ensemble classifier. Traditional machine-learning methods need to extract or generate features from raw sequences or other domain knowledge. Common features include protein physical and chemical properties, water solubility, sequence similarity, position weight matrix, etc. However, these manually designed feature extraction methods rely on an understanding of phosphorylation biological information and may have certain limitations. The prediction performance of machine-learning methods largely depends on the effectiveness of feature extraction, which may result in prediction biases.

Deep-learning models, due to their strong feature representation capabilities, can largely overcome the limitations of traditional feature engineering. Existing deep-learning methods typically involve two steps: First, the protein sequence is divided into windows, and feature embeddings are extracted from the sequence through a deep-learning algorithm, such as convolutional neural networks (CNNs) and recurrent neural networks(RNNs); then, the model is trained to output the results. The classic MusiteDeep framework [33], which utilizes one-hot encoding and multilayer CNNs with attention mechanisms, implements both general and kinase-specific phosphorylation site prediction. Based on MusiteDeep, DeepPhos [34] also employs CNNs and improves performance using multiwindow. Long short-term memory networks (LSTMs) are a type of RNN that can capture the long-term dependencies of sequence tasks. DeepPPSite [35] uses stacked LSTM structures, and DeepPSP [36] utilizes bidirectional LSTMs. DeepIPs [37] uses word embedding methods from NLP to encode protein sequences, combined with CNNs and LSTMs for phosphorylation site prediction.

Protein structures belong to non-Euclidean structured data, so they are extremely complex. CNNs and RNNs cannot address the issue of information retrieval from non-Euclidean structures, thus requiring consideration of other deep-learning neural networks conducive to feature extraction from non-Euclidean data spaces. Therefore, this paper proposed a model that applies graph neural networks to phosphorylation site prediction, treating the protein contact map as an undirected graph data structure. A protein is represented as a topological graph, where nodes are amino acids, and edges represent whether two amino acids are in contact. Additionally, four different types of features are chosen to represent amino acids from different dimensions, including one-hot encoding, physicochemical properties, evolutionary information, and structural information of sequences. By integrating protein sequence and structural information to examine the adjacency relationships between amino acids as well as using graph neural network framework ultimately achieving phosphorylation site prediction.

GraphPhos has two main innovative points: Firstly, the protein structure information is introduced into the model in the form of graphs and processed using the framework of graph neural networks; secondly, GraphPhos is a universal model for predicting amino acid sites of three types: serine, threonine, and tyrosine. Compared with current mainstream prediction models for different types of amino acids, the accuracy has also been significantly improved.

## 2. Results and Discussion

### 2.1. Comparison of General and Specific Category Site-Prediction Performance

Without considering the kinase-specific effects of different amino acid sites, the differences in amino acid residue types are ignored in model training and compared with the results of separate training of different types of amino acids. The comparative results are as shown in Table 1.

The bold data in the table represent the prediction results of all amino acid positions jointly trained and are also the highest accuracy set of experimental results. This indicates that GraphPhos is significantly better at predicting universal amino acid sites than site predictions for specific amino acid types. In terms of accuracy, the model trained using universal sites is improved by about 3%, 6%, and 12.1%, respectively, compared to S, T, and Y sites; the prediction accuracy is improved by about 5.9%, 11%, and 20.6%; F1 score value increased by about 4.5%, 8.8% and 17%.

Since the enzymes that catalyze the phosphorylation of different types of amino acids are different, the characteristics of phosphorylation of different types of amino acids are also different. In other deep-learning methods, in order to obtain better prediction results, S/T and Y amino acid residues are often separated, and the models are trained separately. GraphPhos is a universal amino acid-type phosphorylation site-prediction method that is applicable to S, T, and Y, and the results are better than the individual amino acid-type model. These results suggest that there are common features of phosphorylation at the global scale of proteins that transcend amino acid types and specific kinases and can be captured by the graph neural network framework proposed here.

### 2.2. Analysis of Protein Structure Results from Different Sources

In this study, constructing the contact map requires the use of protein tertiary structures. Since protein tertiary structures may be either known or unknown in practical applications, this section conducts comparative analysis experiments using real structures retrieved from the PDB and predicted structures generated by AlphaFold2 [38]. The experiments include the following scenarios:(i)training with experimental structures and predicting with experimental structures;(ii)training with AlphaFold-predicted structures and predicting with experimental structures;(iii)training with AlphaFold-predicted structures and predicting with AlphaFold-predicted structures.

Table 2 presents the comparative analysis results using protein structures from different sources. The results in the table show that the differences among the three experimental setups are minimal, with the largest variation in Accuracy, Precision, and F1 scores being less than 0.7%. This indicates that even when the true protein structure is unknown, using only predicted protein structures for model training and prediction can still yield satisfactory results, demonstrating that the proposed GraphPhos is applicable to a wide range of real-world scenarios.

These results can be attributed to AlphaFold2’s remarkable ability to predict protein tertiary structures with exceptionally high accuracy, even at the atomic level, as reflected by an average pLDDT (predicted Local Distance Difference Test) value of 68.59 in the training set. To further assess the impact of protein tertiary structure prediction accuracy on the model’s performance, we divided the training set into two subsets based on AlphaFold2’s pLDDT values. The training set was evenly split into two groups: AlphaFold2-high and AlphaFold2-low, with average pLDDT values of 79.188 and 58.956, respectively. The AlphaFold2-predicted structures from the earlier experiments were used as the test set, and the experimental results are shown in Table 3.

The results clearly show that prediction accuracy notably impacts the model’s overall performance, as lower-quality tertiary structure predictions degrade the model’s predictive performance. Observing the results from AlphaFold2-high, the differences compared to the corresponding results in Table 2 (first row) are minimal. This indicates that as long as the training set includes a sufficient number of high-quality predictions, the model’s performance will not be greatly affected and is not sensitive to the average prediction quality of tertiary structures in the training set.

### 2.3. Ablation Study

#### 2.3.1. The Ablation of Modules

In order to test the effectiveness and necessity of the modules used in GraphPhos while ensuring the control variables, two ablation experiments were designed to construct two comparative models without using the graph neural network GraphSAGE module and without using the multilayer convolution module, three models known as GraphPhos, GraphPhos (w/o SAGE), GraphPhos (w/o ProtBERT-CNN). The comparison prediction results of three network structures of various amino acid phosphorylation sites are as Table 4.

Compared with the GraphPhos (w/o SAGE) structure, GraphPhos has improved accuracy and F1-Score by 10% and 4%, respectively. But the accuracy has decreased. Compared with the GraphPhos (w/o ProtBERT-CNN) structure, the accuracy and F1-Score of GraphPhos are increased by 10% and 4.7%, and the accuracy index is also reduced by about 1%. This situation indicates that our proposed model generally improves predictions for negative classes while experiencing a slight decrease in predictions for true positive classes.

The prediction results of the model trained separately for S, T, and Y amino acids are similar to those of universal sites. After reducing the modules, the predictive performance of the model may experience varying degrees of decline. From the four tables, it can be observed that the predictive efficacy of common sites is often the best, followed by S, T, and Y in descending order.

#### 2.3.2. The Ablation of Features

To validate the effectiveness of the feature combinations in this paper, we conducted ablation experiments on each class of features based on all feature combinations, and the predicted results are shown in Table 5.

It can be observed from the results that no matter which feature is ablated, the prediction results obtained are not as good as using all four features. For universal amino acid sites, eliminating any feature will lead to a decrease in accuracy and F1-Score, among which the ablation of physicochemical properties has the greatest impact.

In addition, in order to compare the effectiveness of the four features selected in this article for prediction results, this experiment also conducted separate model training for each type of feature and recorded the prediction results. The test results obtained after training each amino acid node with a single feature were plotted as line graphs individually in Figure 1.

The dotted line in the figure represents the prediction performance obtained by training using all features. The results of the four graphs generally show the same trend. Especially when the physicochemical properties of amino acids and the position-specific scoring matrix are used alone, the prediction results of the model are better than in other cases. It is also noticed that in the process of predicting amino acid sites of all categories, good results were also obtained using only one-hot features. This means that when the sample size is sufficient, the model can learn a good feature representation even by only aggregating the binary encoding information of amino acids adjacent to the structure, which fully illustrates the advantages of aggregating the spatial structure of proteins.

### 2.4. Comparison Results with Other Methods

In order to compare the performance of the models, this paper selected some existing phosphorylation site-prediction models, including machine-learning methods: Musite [25], PhosphoSVM [26] and iPhos-PseEn [32]; deep-learning method: MusiteDeep [33], DeepPhos [34], DeepPSP [36] and DeepIPs [37]. Table 6 shows the prediction accuracy of each model at different amino acid sites.

It is not difficult to find that the accuracy of GraphPhos has been significantly improved on the three amino acid positions. Compared with MusiteDeep, which performs best among the above comparison models, the accuracy of prediction for S, T, and Y sites has increased by 8%, 15%, and 12%, respectively. It is fully demonstrated that the model that applies a graph neural network to the task of predicting phosphorylation sites performs best on the prediction index of Accuracy compared with other methods. It also proves that the model, features used in this model, and the idea of aggregating features from protein space neighbor nodes are very effective and considerable.

### 2.5. Generalizability Analysis

In the previous experiments, to ensure a fair comparison with other methods, we adopted the same dataset-splitting strategy they used, which is a random splitting strategy named GraphPhos-ran. Specifically, the dataset contains 2171 protein sequences, with 1785 sequences randomly assigned to the training set and the remaining 386 sequences to the test set. However, GraphPhos-ran does not account for the potential high similarity between the training and test samples, which may inflate model performance and reduce the credibility of the evaluation. To address this limitation, we conducted an additional set of experiments based on a new splitting strategy named GraphPhos-sim, which considers protein similarity. In simple terms, this strategy divides the training and test sets by protein families, ensuring that samples from the same family do not appear in both sets. Specifically, protein families were first sorted by the number of proteins they contain in descending order. From these, 1785 sequences were assigned to the training set while ensuring that the 386 test sequences came from families completely distinct from those in the training set. This splitting approach ensures that the test set contains samples exclusively from unseen protein families, providing a more rigorous evaluation of the model’s generalization capability. The experimental results are shown in Table 7.

As shown in Table 7, under the random splitting strategy (GraphPhos-ran), the model achieved an Accuracy of 0.9136, a Precision of 0.8566, and an F1-Score of 0.8810. In contrast, under the protein family-based splitting strategy (GraphPhos-sim), the model achieved an Accuracy of 0.8718, a Precision of 0.8932, and an F1-Score of 0.8811. These results indicate that while GraphPhos-ran slightly outperforms GraphPhos-sim in terms of Accuracy, the latter excels in Precision. Notably, the F1 scores for both strategies are nearly identical. This consistent performance across two distinct splitting strategies highlights the model’s strong generalization capability. The nearly identical F1 scores further confirm the model’s ability to maintain a balanced trade-off between Precision and Recall, which is crucial for real-world applications.

## 3. Materials and Methods

### 3.1. Datasets

This paper extracted proteins with phosphorylation reactions from the UniProt protein database [39], with sequence lengths ranging from 500 to 1300, totaling 2171 sequences. Of these, 1785 were used as the training set, and 386 as the test set. The phosphorylation sites of these proteins were annotated. These annotations were derived from manual curation, automated annotation from the UniProt database, as well as annotations from the PRIDE [40] proteomics database.

In the experiment, the amino acid sites with annotation of phosphorylation were defined as positive samples, and the remaining serine (S), threonine (T), and tyrosine (Y) sites were negative samples. During the experiment, an equal number of amino acids were randomly selected from the negative samples to match the number of positive samples for training, thus achieving a balance between positive and negative samples during training.

Phosphorylation is a reaction catalyzed by kinases, and different types of amino acids correspond to different kinases. Therefore, when dividing the dataset, this paper further distinguished S, T, and Y sites. The resulting dataset is shown in Table 8.

### 3.2. Features

The application of protein raw data was from two perspectives: protein sequence and structural features. Protein sequence features were derived partly from manually calculated features commonly used in traditional machine-learning methods and partly from the automatic extraction of amino acid features using pre-trained protein language models(pLMs). Protein structural features were amino acid contact maps that were obtained by converting the protein tertiary structure.

#### 3.2.1. Sequence Features

The protein sequence features include manually calculated features and pLMs (protein language model) features. The manually calculated features consist of four types:


*Manual features:*
One-hot encoding, with a dimension of L×21, where *L* represents the sequence length.Physicochemical properties of proteins, where different types of amino acids are represented by a 5-dimensional principal component vector, resulting in a dimension of L×5 [41].Position-specific scoring matrix (PSSM), calculated based on the SwissProt [42] protein database and generated through sequence alignment using NCBI-blast-2.13.0 [43], with a dimension of L×20.Secondary structure (SS) of proteins, including 8 secondary structure categories: 310-helix (G), α-helix (H), π-helix (I), β-bridge (B), β-strand (E), bend (S), β-turn (T), and coil (C). Additionally, secondary structure features incorporate solvent-accessible surface area (ASA), as well as upper and lower hemisphere exposures (HSEU and HSED), resulting in a total dimension of 11. These features are computed using the SPOT-1D-single method [44].


The final sequence feature dimension is L×57, and the concatenation method is shown as(1)Features=[onehot,phy,pssm,ss]

The combined features of every amino acid are finally represented as nodes in the protein contact graph *G*, which have *L* nodes, each representing an amino acid molecule in the protein sequence.

*Pre-trained protein language model features:* This paper used ProtTrans [45], more specifically ProtBERT, to extract protein sequence features and took its encoder output as our model input features. To comply with other features, the embedding dimension is projected from 1024 dimensions to 57 dimensions by two convolution layers.

#### 3.2.2. Structural Features

This study utilized experimentally verified protein tertiary structures, as well as protein tertiary structures predicted by AlphaFold2 [38]. To obtain the actual protein structures, we first retrieve specific protein IDs through the UniProt [39] database and then use these IDs to conduct precise searches in the Protein Data Bank (PDB) to acquire the corresponding structural information. The predicted tertiary structures were retrieved from the AlphaFold2 protein structure database. AlphaFold2 is a leading computational method in the field of protein structure prediction and is able to achieve structure prediction at the atomic level.

The tertiary structure of proteins is represented by PDB files, and subsequently, contact maps are calculated to reflect the contact information and structural characteristics between amino acids. These contact maps serve as the protein structural features used in this paper. The input graph structure of the model is the topological representation of the amino acid contact map of the protein. The method proposed by Godzik et al. in 1994 [46] was used to calculate the contact map for each protein from its PDB file. For a protein sequence of length *L*, 1≤i,j≤L, its amino acid nodes are $v=\{v_{1},\ldots ,v_{i},\ldots ,v_{L}\}$. In space, the three-dimensional coordinates of amino acids are represented as $\left(x_{v_{i}},y_{v_{i}},z_{v_{i}}\right)$. Therefore, the calculation of Euclidean distance between amino acids is shown as(2)$$Dist\left(C_{\beta_{i}},C_{\beta_{j}}\right)=\sqrt{{\left(x_{v_{i}}-x_{v_{j}}\right)}^{2}+{\left(y_{v_{i}}-y_{v_{j}}\right)}^{2}+{\left(z_{v_{i}}-z_{v_{j}}\right)}^{2}}$$
where $C_{\beta }$ refers to the β-C atom, distance matrix Dist of L×L dimensions between amino acids can be obtained, where each entry in the matrix represents the distance between amino acids, i.e., ${Dist}_{ij}=Distance\left(C_{\beta_{i}},C_{\beta_{j}}\right)$. If the Euclidean distance between two amino acids ${Dist}_{ij}\leq 8\overset{\circ }{A}$, it is considered that there is contact between these two amino acids.

By combining protein sequence features with structural features, they serve as the model’s input, enabling the joint guidance of protein sequence and spatial structure in predicting protein-phosphorylation sites.

### 3.3. Methods

This paper proposed an algorithm based on GraphSAGE and ProtBERT, which consists of three modules, namely the GraphSAGE module. ProtBERT combines a multilayer CNN module and a multilayer FNN module, respectively. The algorithm framework is shown in Figure 2.

Using the feature extraction method described in the previous section, a manual feature matrix $X\in R^{L\times f}$ of length *L* protein sequence can be obtained, *f* means that each amino acid node has a *f* dimensional feature vector. In the protein contact map, each graph node corresponds to an amino acid, and the feature of each node corresponds to the feature of each amino acid. Whether there is an edge between two nodes in the graph corresponds to whether there is contact between two residues in the contact matrix. The artificial feature matrix and the graph representation jointly serve as the input for GraphSAGE.

#### 3.3.1. GraphSAGE

GraphSAGE [47] is a graph neural network model proposed by Hamilton et al. in 2017. This model improves upon GCN [48], a classic graph neural network model introduced in 2016. GCN introduces a graph convolutional layer based on the Laplacian matrix, which aggregates neighbor information of nodes into their representations through normalization and propagation of the adjacency matrix within the framework of convolutional neural networks. This graph convolution operation allows neural networks to perform convolutions on nodes, enabling the representations of nodes to capture the structural and semantic information of their neighbors, facilitating simple and effective processing of graph-structured data. GraphSAGE improves upon GCN in two main aspects: sampling strategy and neighborhood aggregation. Instead of sampling the entire graph as in GCN, GraphSAGE optimizes sampling to focus on partial neighboring nodes, sampling neighbor subgraphs, and aggregating neighbor node information. This allows the model to learn and generalize to unseen nodes during training, thereby possessing inductive learning characteristics. Such sampling strategies are more conducive to training large-scale graph data. GraphSAGE also explores various neighborhood aggregation methods and compares the advantages and disadvantages of different methods. The flow of GraphSAGE sampling and aggregation methods is illustrated in Figure 3.

The algorithm process of GraphSAGE is roughly divided into three steps: First, sampling of nodes. Figure 3a illustrates the sampling of two-hop neighbors, with two neighbors sampled at each hop. If the number of neighbors of the central node exceeds the sampling quantity, random selection is made from the neighbors; otherwise, if there are fewer neighbors than the sampling data, repeated sampling is required. Figure 3b shows the aggregation process of node neighbors, where different aggregation methods can be employed at each step. Finally, in Figure 3c, other downstream tasks are executed based on the aggregated node information learned.

Define the input graph G(V,E) and input feature $\left\{x_{v},\forall v\in V\right\}$ the initialization is shown as(3)$$h_{v}^{0}\leftarrow x_{v},\forall v\in V$$
the neighbor aggregation is obtained as(4)$$h_{N(v)}^{k}\leftarrow {AGGREGATE}_{k}\left(\left\{h_{u}^{k-1},\forall u\in N(v)\right\}\right)$$
where *k* represents the order of sampled neighbor nodes,∀k∈{1,…,K} and *K* represents the maximum order of neighbors that can be sampled, i.e., the maximum depth that can be traversed starting from any node. $N:v\to 2^{V}$ represents the neighbors of the sampling of node *v*. Then, concatenate the aggregation of each node with the features of upper-order neighbor aggregation as(5)$$h_{v}^{k}\leftarrow \sigma \left(W^{k}\cdot CONCAT\left(h_{v}^{k-1},h_{N(v)}^{k}\right)\right)$$

Finally, the aggregated node features are regularized(6)$$h_{v}^{k}\leftarrow h_{v}^{k}\setminus {∥h_{v}^{k}∥}_{2}$$

${∥h_{v}^{k}∥}_{2}$ represents the norm of features, i.e., calculating the Euclidean distance between features. After completing *K* loops, the final vector representation is obtained as(7)$$z_{v}\leftarrow h_{v}^{K},\forall v\in V$$

GraphPhos uses two layers of GraphSAGE to aggregate second-order neighbors. Each layer of GraphSAGE can be expressed as the following formula(8)$$X_{vi}^{l}=\sigma \left(GMEAN\left(X_{vj}^{l+1}\right)W^{l+1}\right),\forall v_{j}\in Neighbor(v_{i})$$
where $X_{l}\in R^{L\times f}$ is the hidden feature of the *l* layer, and the initial dimension is L×57, $W^{l}\in R_{f\times f^{\prime }}$ is a trainable weight matrix that maps the initial feature matrix from *f* dimension to $f^{\prime }$, and σ represents a nonlinear activation function, MEAN indicates the use of average aggregation. This model uses ReLU as the activation function and adds a normalization layer after each GraphSAGE layer to map the output to [0,1], which can reduce training time, alleviate gradient disappearance and explosion problems, and accelerate model convergence. Finally, a two-layer GraphSAGE graph neural network is used to learn the characteristics of the central node, and the output is shown as(9)$$X_{graphsage}=ReLU\left(GReLU\left({GX}_{seq}W^{\left(1\right)}\right)W^{\left(0\right)}\right)$$

#### 3.3.2. ProtBERT Combines multiCNN

As an NLP pretraining model, Bert has achieved great success on many tasks. Elnaggar A et al. extended Bert to protein representation by changing the training data sets from natural language corpus to large protein sequences data sets [45]. In order to maximize memory usage, ProtBert adds a few layers based on the original Bert architecture. The model had been trained on two protein data sets: UniRef100 [49] and BFD100 [50], respectively. BFD100 is created by fusing UniProt [39] with numerous metagenomic sequencing studies, removing duplicate data and integrating the results, thereby forming the largest collection of protein sequences. ProtBert–UniRef was first trained for 300 k steps on a maximum-length 512 sequence and then for an additional 100 k steps on a maximum-length 2 k sequence. On the other hand, ProtBert-BFD trains the sequences for 800 k steps and then 200 k steps, respectively, with maximum lengths of 512 and 2 k. In this way, the model first extracted useful features from shorter sequences, and then, by taking large batch sizes, it can train longer sequences more efficiently.

The framework of ProtBert is shown in Figure 4. Denote the input protein sequence as X, then it is embedded by integrating amino acid embeddings, segment embeddings, and position embeddings as(10)$$X_{emb}=Lookup\left(X\right)+Segment\left(X\right)+Positional\left(X\right)$$

Following this, the embedding $X_{emb}$ is further refined by passing it through the ProtBert model to generate the encoded representation $X_{encoder}$. This process is encapsulated in the equation:(11)$$X_{encoder}=\mathrm{ProtBert}\left(X_{emb}\right)$$

After calculation by ProtBERT, the protein sequences are mapped to a high-dimensional feature space to obtain the feature representation for each protein, denoted as $F\in R^{L\times p}$, *p* is a fixed dimension output of ProtBERT. In order to jointly guide the prediction of protein-phosphorylation sites with the features processed by the graph neural network mentioned above, it is necessary to reduce the dimensionality of feature *F*. Additionally, feature *F* represents general features extracted by ProtBERT for amino acids and should undergo further processing to adapt to the downstream task of phosphorylation site prediction.

This paper uses a multilayer convolutional neural network to further extract ProtBERT features. The input is the feature representation $F\in R^{L\times p}$. The calculation formula of each convolutional layer of the 1-dimension convolutional neural network used in this experiment is as follows:(12)$$F_{l+1}=Conv1d\left(WF_{l}+b\right)$$

$F_{l}$ and $F_{l+1}$ are the input and output features that pass through the *l*th convolutional layer, respectively. *W* corresponds to the weight matrix of the convolutional layer. In the multilayer CNN module, normalization operations are also added to each convolution block, and ReLU is used as the activation function. The architecture of multilayer convolution layers has been shown in Figure 5.

This experiment sets the number of layers of CNN to 2, and the final output of the MultiCNN module is(13)$$F_{multicnn}=ReLU\left(ReLU\left(F_{seq}W^{\left(0\right)}+b^{\left(0\right)}\right)W^{\left(1\right)}+b^{\left(1\right)}\right)$$

#### 3.3.3. MultiFNN

The outputs of the two modules $X_{graphsage}$ and $F_{multicnn}$ described above are concatenated to obtain the feature representation $M\in R_{L^{\prime }\times f}$ of the amino acid nodes of interest in the protein sequence, expressed as(14)$$M=[X_{graphsage},F_{multicnn}]$$
where $L^{\prime }$ is the number of amino acid nodes of interest in the protein sequence. During the aggregation and computation processes of the two modules mentioned above, focusing only on the S, T, and Y amino acid positions. Therefore, the filtered feature dimension is $L^{\prime }$. The concatenated feature representation $M\in R^{L^{\prime }\times f}$ is then inputted into a classification module composed of fully connected layers to predict the probability of protein amino acid phosphorylation.

This module employs two fully connected layers. The first fully connected layer converts the dimension *f* into the hidden layer dimension *h*, followed by another fully connected layer to obtain the output. Since this experiment is a binary classification task, the Sigmoid function is used to calculate the probability of an amino acid being phosphorylated, as follows:(15)$$P=Sigmoid(W_{2}\left(W_{1}M\right))$$
where $W_{1}$ and $W_{2}$ are the weight matrices of the two fully connected layers. When P⩾0.5, the position is considered phosphorylated; otherwise, is not.

### 3.4. Implementation Details

GraphPhos was implemented with pytorch (version 1.13.1) [51] and trained on NVIDIA RTX 3090 24G (Santa Clara, CA, USA). The initial learning rate is $10^{-3}$. Adam optimizer was used to update network parameters. In this experiment, adding normalization layers between each layer is beneficial.

### 3.5. Evaluating Indicator

In order to evaluate the phosphorylation site-prediction performance of GraphPhos, several common binary classification evaluation indicators were applied: Accuracy (Acc), Precision (Pre), and F1-Score. Defined as follows:(16)$$Acc=\frac{TP+TN}{TP+TN+FP+FN}$$(17)$$Pre=\frac{TP}{TP+FP}$$(18)$$F1-Score=\frac{2\times Pre\times Sn}{Pre+Sn}$$
where Sn indicates the proportion of all actual positive samples that are correctly predicted as positive(19)$$Sn=\frac{TP}{TP+FN}$$TP, TN, FP, FN are the number of categories from the confusion matrix.

## 4. Conclusions

This paper proposes a protein-phosphorylation site-prediction method based on graph neural networks, GraphPhos. Not only does it successfully innovate the application of graph neural networks, but it also incorporates actual protein secondary structure information into the model. The experiment results demonstrate remarkable performance in terms of Accuracy, Precision, and F1-Score, and GraphPhos performs better in predicting phosphorylation sites without distinguishing specific amino acid types.

In recent years, biotechnology has made continuous breakthroughs, leading to the rapid accumulation of phosphorylation data. However, many upstream kinases catalyzing phosphorylation events remain unknown. Being able to identify phosphorylation sites and determine their homologous kinases at the same time, achieving comprehensive prediction of kinases and substrates will provide more powerful biological annotations at the proteome level.

## Figures and Tables

**Figure 1 ijms-26-00941-f001:**
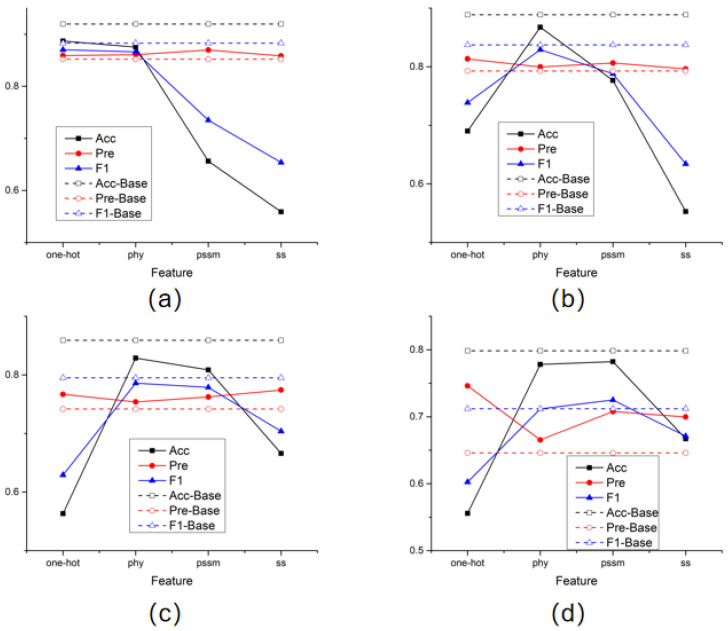
Single feature training results. (**a**) the results of STY amino acids; (**b**) the results of S amino acids; (**c**) the results of T amino acids; (**d**) the results of Y amino acids.

**Figure 2 ijms-26-00941-f002:**
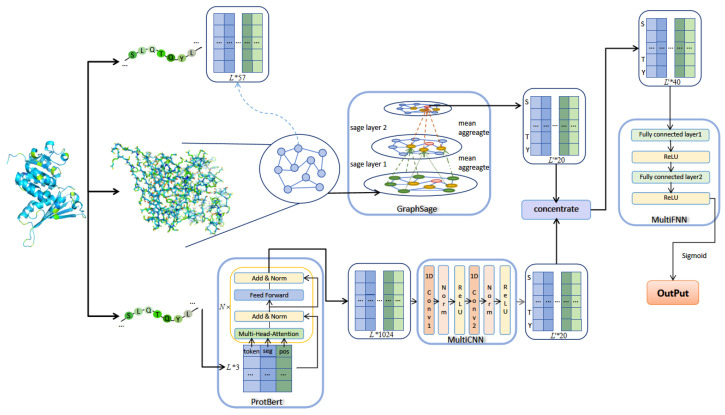
The architecture of GraphPhos.

**Figure 3 ijms-26-00941-f003:**
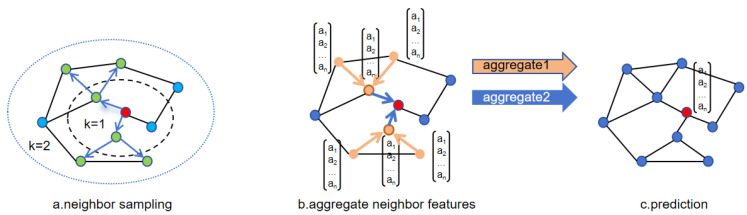
Sampling and aggregation of GraphSAGE.

**Figure 4 ijms-26-00941-f004:**
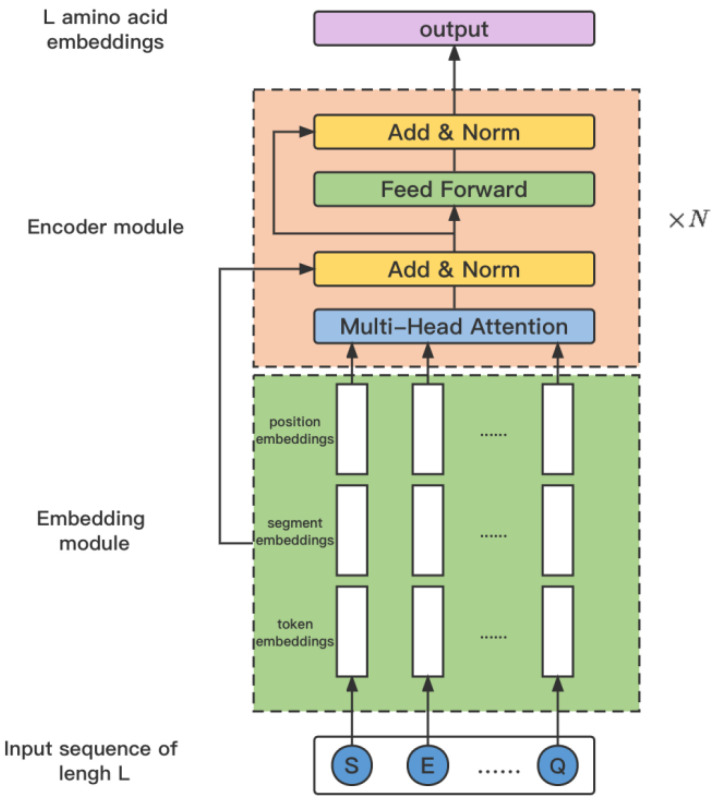
The architecture of the model ProtBERT.

**Figure 5 ijms-26-00941-f005:**
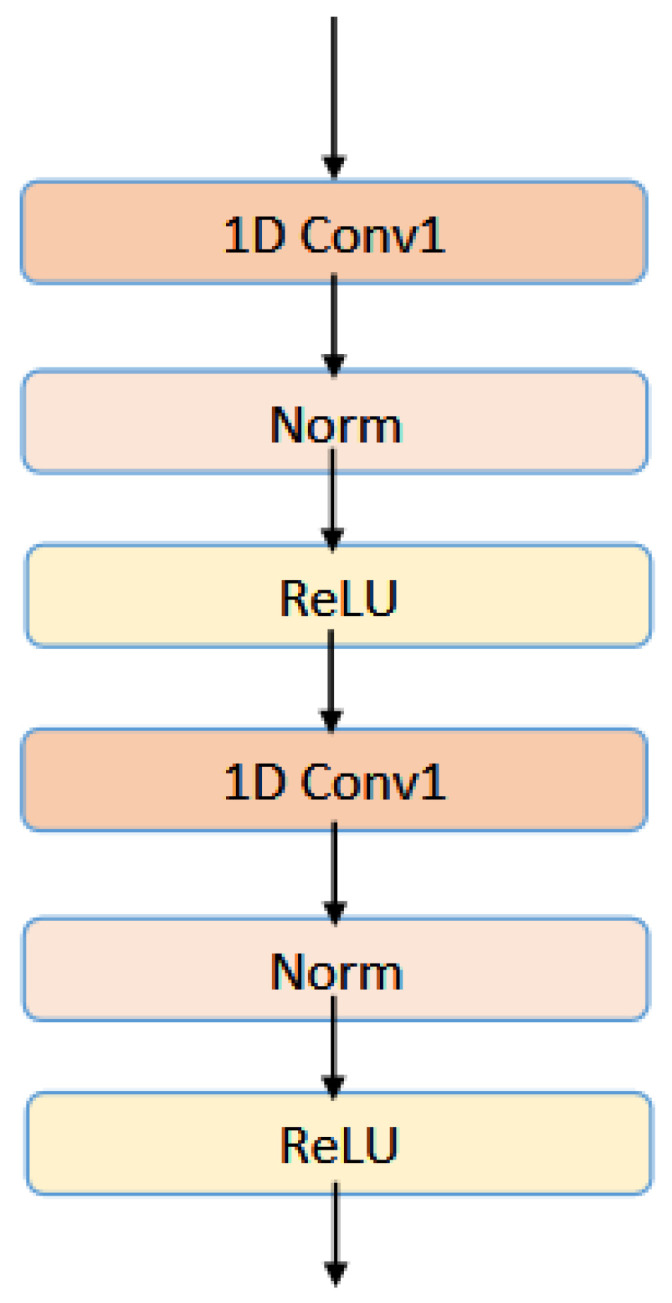
The architecture of multilayer convolution layers.

**Table 1 ijms-26-00941-t001:** Experimental results of different types of amino acids.

	Accuracy	Precision	F1 Score
S	0.8888	0.7927	0.8374
T	0.8590	0.7418	0.7951
Y	0.7985	0.6460	0.7120
STY	**0.9196**	**0.8519**	**0.8829**

**Table 2 ijms-26-00941-t002:** Prediction results of protein structures from different sources.

Train	Test	Accuracy	Precision	F1-Score
AlphaFold2	AlphaFold2	0.9136	**0.8566**	0.8810
PDB	AlphaFold2	**0.9197**	0.8509	0.8828
	PDB	0.9196	0.8519	**0.8829**

**Table 3 ijms-26-00941-t003:** Prediction results of protein structures from different quality prediction models.

Train	Test	Accuracy	Precision	F1-Score
AlphaFold2-high	AlphaFold2	0.9174	0.8543	0.8823
AlphaFold2-low	AlphaFold2	0.4244	0.8445	0.5133

**Table 4 ijms-26-00941-t004:** Module ablation experiment.

	Accuracy	Precision	F1-Score	
STY	GraphPhos	**0.9196 **	0.8519	**0.8829**
	GraphPhos (w/o SAGE)	0.8271	0.8589	0.8398
	GraphPhos (w/o ProtBERT-CNN)	0.8277	**0.8606**	0.8408
S	GraphPhos	**0.8888**	0.7927	**0.8374**
	GraphPhos (w/o SAGE)	0.6524	0.8036	0.6999
	GraphPhos (w/o ProtBERT-CNN)	0.8593	**0.8054**	0.8271
T	GraphPhos	**0.8593**	**0.8054**	**0.8271**
	GraphPhos (w/o SAGE)	0.7371	0.7586	0.7425
	GraphPhos (w/o ProtBERT-CNN)	0.8590	0.7418	0.7951
Y	GraphPhos	**0.7985**	0.646	0.712
	GraphPhos (w/o SAGE)	0.7851	**0.6665**	**0.7144**
	GraphPhos (w/o ProtBERT-CNN)	0.7516	0.6387	0.6878

**Table 5 ijms-26-00941-t005:** Feature combination ablation experiment.

		Accuracy	Precision	F1-Score
phy pssm ss	S	0.8729	0.8042	0.8332
	T	0.8034	0.7577	0.7763
	Y	0.7237	0.7023	0.7017
	STY	**0.9084**	**0.8563**	**0.8790**
one-hot pssm ss	S	0.6712	0.7966	0.7133
	T	**0.8539**	0.753	**0.7951**
	Y	0.6598	0.6893	0.6696
	STY	0.6339	**0.8606**	0.7161
one-hot phy ss	S	**0.8671**	0.8086	**0.8328**
	T	0.7247	0.7642	0.7395
	Y	0.5237	0.6911	0.5517
	STY	0.8017	**0.8511**	0.8223
one-hot phy pssm	S	0.8766	0.7987	0.833
	T	0.8574	0.7616	0.7985
	Y	0.7907	0.6515	0.7099
	STY	**0.8831**	**0.8537**	**0.8661**

**Table 6 ijms-26-00941-t006:** Comparison of Accuracy between models at different amino acid positions.

	S	T	Y
Musite	0.6923	0.6478	0.6180
PhosphoSVM	0.7086	0.6669	0.6391
iPhos-PseEn	0.7680	0.7520	0.7900
MusiteDeep	0.8095	0.8095	0.8571
DeepPhos	0.6890	0.6890	0.6890
DeepPSP	0.8021	0.8021	0.7619
DeepIPs	0.8063	0.8063	0.8333
**GraphPhos**	**0.8819**	**0.9600**	**0.9779**

**Table 7 ijms-26-00941-t007:** The results of different selection strategies on the dataset.

	Accuracy	Precision	F1-Score
GraphPhos-ran	**0.9136**	0.8566	0.8810
GraphPhos-sim	0.8718	**0.8932**	**0.8811**

**Table 8 ijms-26-00941-t008:** Datasets.

	Train Set		Test Set	
	Positive	Negative	Positive	Negative
Serine	10,420	10,420	2814	2814
Threonine	1244	1244	224	224
Tyrosine	695	695	80	80

## Data Availability

The source code of GraphPhos is publicly deposited in https://github.com/zezemua/GraphPhos (accessed on 26 November 2024).

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
