# Peer review of "GraphPhos: Predict Protein-Phosphorylation Sites Based on Graph Neural Networks"

_ijms, 2025, doi:10.3390/ijms26030941_

Round 1
Reviewer 1 Report
Comments and Suggestions for Authors
The authors present a novel method for the prediction of phosphorylation sites. The method adopts a complex network and seems to achieve good performance, however, I have some major concerns regarding the use of data:
- The authors report the number of residues included in the dataset but I did not see mention of how many proteins they come from. Similarly, the authors should report how many proteins have experimental structures and how many only have the AlphaFold predicted structure.
- The authors do not mention how data was split into training and testing datasets. It is important to separate proteins used for training and those used for testing making sure that no protein in the test set is similar to any protein seen during training. This is needed to avoid overfitting and to ensure that the reported metrics are somewhat representative of the real behavior of the method. Moreover, all decisions regarding the model architecture and hyperparameters should be made in cross-validation on the training set, using the test set only for the final benchmark.
- The authors perform an experiment to validate the adoption of AlphaFold structures during training. This is acceptable if the predicted models have good quality (the authors should report a statistic on the quality of the adopted models), however, I would expect that the performances of the model would drastically drop when using low-quality models as input. Would it be possible to perform a study to reflect how the performances change at varying levels of pLDDT of AlphaFold models used in input?
I also have a few minor concerns regarding the manuscript:
- The authors mention that they get protein models from UniProt, but this is not true. UniProt stores protein sequences and it has cross-links to PDB (as well as to AlphaFoldDB). The manuscript should mention that experimental structures are retrieved from the PDB.
- I would move the definition of evaluation metrics in the materials and methods section.
- An extensive description of ProtBERT is in my opinion not needed and could be simply deferred to the original paper.
Reviewer 2 Report
Comments and Suggestions for Authors
The authors presented GraphPhos, a novel phosphorylation site prediction method that combines a protein language model with a graph neural network. The method achieved better performance compared to existing methods. The authors provide a clear explanation of their methodology, supported by ablation analysis. While the study is impressive, I have a few minor comments that can be addressed with a minor revision.
Additionally, there are some inaccuracies in the use of structural biology terms. The authors may not be fully familiar with the protein structure field. I recommend asking a structural biologist to refine these aspects for improved clarity.
Minor:
1. Use of “secondary”. In the abstract, “the secondary structure”, in section 2.2.2 “a secondary contact map”, and in section 5 “protein secondary structure information”. There are several “secondary” in the manuscript, but the usage of the term “secondary” is not correct. The term “secondary structure” refers specifically to local structural motifs such as alpha-helices, beta-sheets, and loops. A “contact map” represents pairwise residue contacts, which is not a secondary structure. Please carefully review the term “secondary” in the manuscript.
2. In 2.2.2 Sequence features: The manual computation of features is not well-explained. For example, how were protein secondary structure features computed? Which specific method or tool was used to generate the PSSM? Please provide a detailed explanation or cite relevant references.
3. In 2.2.2 Structural Features: Please explain what is Cb (maybe beta-carbon in the amino acid).
4. In 4.3, Does "real structure" refer to experimentally determined protein structures available in the PDB? If so, please specify this. I think the number of available protein structures in PDB is very limited. Please also provide the list of proteins used in the training and test datasets.
5. The code is not available in GitHub.
Round 2
Reviewer 1 Report
Comments and Suggestions for Authors
The authors addressed most of my concerns; however, they confirmed that the training and testing datasets were randomly split without consideration of protein similarity. The acknowledgment that previous studies did not follow good practices that are established nowadays is in my opinion no excuse to perpetrate this. If anything, pointing out the flaws of previous experiments would be a good reason to develop a new state-of-the-art (even if the performances would end up being lower), as I would not trust anyway the performances of a method that was tested on proteins similar to those seen during training.
I suggest properly splitting the dataset and repeating the training procedure of the final architecture, thus reporting performances that would better reflect the expected quality of the method.
Round 3
Reviewer 1 Report
Comments and Suggestions for Authors
The authors proved that their model can show good generalization capabilities under rigorous splitting of data, improving the soundness of their results.